# Strategically-timed State-Observation Attacks on Deep Reinforcement Learning Agents

**You Qiaoben** [1]   **Xinning Zhou** [1]   **Chengyang Ying** [1]   **Jun Zhu** [1]

## Abstract

Deep reinforcement learning (DRL) policies are vulnerable to the adversarial attack on their observations, which may mislead real-world RL agents to catastrophic failures. Several works have shown the effectiveness of this type of adversarial attacks. But these adversaries are inclined to be detected because these adversaries do not inhibit their attacks activity. Recent works provide heuristic methods by attacking the victim agent at a small subset of time steps, but it aims at lack for theoretical principles. Inspired by the idea that adversarial attacks at each time step have different efforts, we denote a novel strategically-timed attack called Tentative Frame Attack for continuous control environments. We further propose a theoretical framework of finding optimal frame attack. Following this framework, we trained the frame attack strategy online with the victim agents and a fixed adversary. The empirical results show that our adversaries achieve the state-of-the-art performance on DRL agents which outperforms the full-timed attack.

## 1. Introduction

Deep reinforcement learning (DRL) agents have achieved outstanding performance in Atari games (Mnih et al., 2016), Go (Silver et al., 2016), and other different challenging tasks (Schrittwieser et al., 2020). However, contemporary studies (Huang et al., 2017; Zhang et al., 2020; 2021) have shown that DRL agents are vulnerable to the adversarial attack and may cause catastrophic failures. Therefore it's essential to investigate the malicious adversarial attack on DRL agents before deploying them to the safety applications such as self-driving.

Since the influential work (Mandlekar et al., 2017) studies the adversarial attack on policies, the adversarial vulnerabilities in DRL agents have been broadly studied. Most of the adversarial attacks (Huang et al., 2017; Pattanaik et al., 2017; Lin et al., 2020) on DRL agents study the adding perturbations on the observations of the victim agent. Although these attacks have shown the effectiveness of deceiving DRL agent, we raise a question: *With a fixed function generating perturbation, is it the strongest adversary to attack the policy at every time step?*

In a recent work, Lin et al. (2017) proposes an adversarial attack at selective time steps instead of attacking at every time step. Since the adversarial attacks at different time steps are not equally effective, this attack reduces the accumulated reward with fewer adversarial perturbations. Although it's proved to be effective in discrete action space, it is not available in continuous control tasks. Sun et al. (2020) proposes a stealthy attack in continuous control tasks but it doesn't consider the ability of the adversary. Inspired by the analysis of Lin et al. (2017), we design a novel strategically-timed attack combined with state-of-the-art adversarial attack in continuous tasks named Tentative Frame Attack. Noting that our algorithm only learns a strategically-timed attack function to decide whether the adversary should apply the perturbation at each state with a fixed function or not. It is different from the setting which learns both victim policy and the perturbation at each state.

However, these approaches do not consider the delayed reward mechanism and lack for theoretical principles. We formulate the strategically-timed attack on state observations as a Markov decision process (MDP), which we call Strategically-timed State-adversarial MDP (SS-MDP). With a fixed preliminary adversary and victim policies, we demonstrate the existence of the optimal frame attack strategy on deciding whether the victim should be attacked at each state. We provide a novel empirical adversarial attack to DRL agents following SS-MDP framework and achieving significantly stronger performances than any other strategically-timed attacks.

To summarize, we study the theoretical and the practical

[1]Dept. of Comp. Sci. & Tech., Institute for AI, BNRist Center, Tsinghua-Bosch Joint ML Center, Tsinghua University, Beijing, China. Correspondence to: You Qiaoben <qby17@mails.tsinghua.edu.cn>.

*Accepted by the ICML 2021 workshop on A Blessing in Disguise: The Prospects and Perils of Adversarial Machine Learning.* Copyright 2021 by the author(s).

properties of strategically-timed attacks on perturbing the state observations of DRL agents with a fixed preliminary adversary. Following the idea that the adversary's effect is depend on the observations, we design a novel heuristic approach in MuJoCo environments with the state-of-the-art adversary. Then we formulate the adversary attack on state observations with a preliminary adversary and a strategically-timed attack function as Strategically-timed State-adversarial MDP (SS-MDP). Based on the SS-MDP framework, we demonstrate the existence of the optimal strategically-timed attack function when fixing the perturbation and the victim policy and further train an adversary online following this framework. We evaluate our approaches on three environments in MuJoCo environments. The empirical results show our strategy following SS-MDP framework outperforms the heuristic strategy and the full-timed attack.

## 2. Related works

In this section, we concisely review the contemporary adversary (Huang et al., 2017; Kos & Song, 2017; Behzadan & Munir, 2017) on perturbing state observations of the DRL models and discuss the divergence between our method and strategically-timed attack (Lin et al., 2017).

### 2.1. Adversarial attack on state observations

Since the adversary is incapable of directly modifying the state in the environment such as in Atari Games (Mnih et al., 2016) or autonomous driving (Dosovitskiy et al., 2017), most adversary preferably perturb the observation received from the environment. Particularly, the existent adversary (Huang et al., 2017; Kos & Song, 2017; Behzadan & Munir, 2017; Pattanaik et al., 2017; Zhang et al., 2020; 2021) generating the adversarial noise and deceive the agent to take a sub-optimal action. Several works (Huang et al., 2017; Pattanaik et al., 2017; Zhang et al., 2020) follow the adversarial robustness studies in supervised learning and conduct the adversarial perturbation with gradient based methods. Other work (Zhang et al., 2021) trains the adversary online together with fixed victim policy and achieves state-of-the-art performance on continues control tasks comparing with other gradient based methods. In our experiment, we choose the state-of-the-art algorithm (Zhang et al., 2021) as our fixed adversary because it significantly outperforms any other algorithms in MuJoCo tasks.

### 2.2. Strategically-timed attack

We briefly review strategically-timed attack (Lin et al., 2017) and indicate the discrepancies between this work and our work. Although both work reduce the accumulated reward of the attacked policy, the previous strategically-timed attack proposes method to solving when and how to attack the victim policy. Sun et al. (2020) proposes a powerful

and stealthy attack without the limitation on the perturbation. Yang et al. (2020) provides a strategically-timed attack by evolutionary strategy with a fixed noise, but lack for theoretical principles. We define a framework for training strategically-timed attack function in section 3.2 and show the existence of the optimal strategically-timed attack under fixed noise. We demonstrate the adversary learned under this framework is stronger than the full-timed attack.

## 3. Methodology

In this section, we propose two strong adversarial attacks for continuous control tasks. We first introduce a strategically-timed attack for continuous control tasks and determine when to attack by a inspired by Lin et al. (2017).

### 3.1. Tentative Frame Attack

In this section, we propose a novel adversarial attack for continuous tasks. As existing method (Lin et al., 2017) is not available for the adversary the least preferred action, we introduce a tentative frame attack for deciding when to attack. The approach is tentative because the adversary compute the tentative function $t$ which represents the original action's Q-values functions over the attacked action's Q-values functions at current states. For policy gradient methods, we use robust policy against small perturbations (Zhang et al., 2020) to provide a better estimation on Q-values functions. Based on this intuition, we define the tentative function $t$ as:

$$t(s) = Q'(s, \pi(s)) - Q'(s, \pi(h(s))),$$

where $\pi$ is the victim policy, $Q'$ is the estimation of Q-values function which denotes the Q-values function of Robust Sarsa, and $h$ denotes the fixed adversary. We choose the optimal attack in Zhang et al. (2021) as applied noise since it's a state-of-the-art attack in continuous control tasks. In our tentative frame attack, we add adversarial noise to the agent's observation when function $t(s)$ exceeds given threshold $\beta$:

$$t(s) > \beta, \tag{1}$$

where $\beta$ is a task-dependent hand-craft hyper-parameter.

### 3.2. Strategically-timed State-observation Markov decision process framework

We follow the setting of SA-MDP framework (Zhang et al., 2020) and define a Strategically-timed State-observation Markov decision process (SS-MDP) framework. The SS-MDP is a 6-tuple $M = (S, A, P, R, \gamma, h)$, where $S$ is the state set of the environment, and $A$ is the action set, and $P : S \times A \to \mathcal{F}(S)$ is the transition function of the environment where $\mathcal{F}(S)$ is the set of all possible probability distributions on $S$, and $R : S \times A \times S \to \mathbb{R}$ is the reward function, and $\gamma$ is a discount factor. The victim policy is defined as

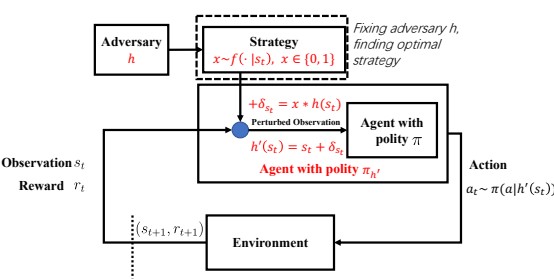

*Figure 1.* Strategically-timed adversarial attacks on the victim's observations: Given an agent with policy $\pi$, the attacker strategically-timed applies the adversary function $h$ to the observation. Specifically, the adversary perturbs the agent based on $x$ sampled from the strategically-timed attack function $f$, and add perturbations when $x$ equals to 1: $h'(s) = s_t + x * h(s_t)$. Then the victim agent consequentially behaves as $a_t \sim \pi(a|h'(s))$, which can be recognized as an agent with strategically-timed attack $a_t \sim \pi'_h(a|s_t)$.

$\pi : S \to \mathcal{F}(A)$, where $\mathcal{F}(A)$ is the set of all distributions on $A$. In SS-MDP, the adversary decides when to perturb the observation with the preliminary adversary $h : S \to \mathcal{F}(S)$ by using $x \sim f(\cdot|s), x \in \{0, 1\}$. Specifically, the adversary perturb the agent if and only if $x$ equals to 1.

Fixing the victim policy $\pi$, the adversary aims to minimize the expected total reward of $\pi$ by strategically-timed applying the perturbations generated by the adversary $h$. In SS-MDP in Fig. 1, we discuss how to find an optimal adversary $h'$ with the given adversary $h$, noting that the given adversary generates the perturbation $h(s)$. In each state, the adversary $h'$ either keeps the observation or perturbs the generated by the adversary, which satisfies: $h'(s) \in \{h(s), s\}$. We denote the strategically-timed attack function $f(s) : S \to [0, 1]$ as the probability of the adversary attacks at each state. Then the attacked policy satisfies:

$$\pi_{h'}(s) \triangleq (1 - f(s))\pi(h(s)) + f(s)\pi(s). \quad (2)$$

With the notation of $\pi_{h'}$, the goal of SS-MDP is to minimize the expected total reward as:

$$h^* = \arg\min_h \mathbb{E}_{a_t \sim \pi_h(.|s_t), s_{t+1} \sim P(s_t, a_t)} \left[\sum_{t=0}^{\infty} \gamma^t r_t\right]. \quad (3)$$

We denote $a \sim \pi_h$ instead of $a_t \sim \pi(.|h(s_t))$ and omit the transition $s_{t+1} \sim P(s_t, a_t)$ in the following part of this paper to simplify the notations.

We can derive the adversarial perturbation by as to maximize

the accumulated reward of $\pi_{h'}$:

$$f^* = \arg\min_f \left[R(\pi_{h'}) \triangleq \mathbb{E}_{a \sim \pi_{h'}} \sum_{t=0}^{\infty} \gamma^t r_t\right]. \quad (4)$$

From the adversary's point of view, we can redefine the victim policy and the environment dynamics as a new MDP and provide a framework to solving problem (4). Inspired by Zhang et al. (2020), we redefine an MDP with two-dimension action space by merging the fixed victim policy, the environment dynamics and the fixed and determined adversary $h$. Then we propose the lemma as below:

**Lemma 1.** *Given an SS-MDP $M = (S, A, P, R, \gamma, h)$, a fixed and determined policy $\pi(\cdot|\cdot)$ and a fixed adversary $h$, there exists an MDP $\hat{M} = (S, \hat{A}, \hat{R}, \hat{p}, \gamma)$ such that the optimal policy of $\hat{M}$ is the optimal strategically-timed attack function $f$ of the adversary for SA-MDP given the fixed $\pi$, where $\hat{A}$ is $\hat{A} = \{0, 1\}$, and the redefined reward satisfies:*

$$\hat{R}(s, \hat{a}, s') := \mathbb{E}\left[\hat{r}|s, \hat{a}, s'\right] = -\frac{\sum_a p(s'|s, a)R(s, a, s')}{\sum_a p(s'|s, a)},$$

*where the redefined system dynamic satisfies:*

$$\hat{p}(s'|s, \hat{a}) = \begin{cases} p(s'|s, \pi_h(s)) & \hat{a} = 0 \\ p(s'|s, \pi(s)) & \hat{a} = 1 \end{cases}$$

The proof can be derived similar to Lemma 1 in Zhang et al. (2021). Since the optimal policy always exists in any MDP (Puterman, 2014), there always exists an optimal strategically-timed attack function. So we model our attack function as a neural network and trained the strategically-timed attack function by PPO.

## 4. Empirical results

In this section, we show extensive experiments by attacking agents trained by PPO (Schulman et al., 2017) based on our methods and baselines in MuJoCo, which demonstrate the effectiveness of our methods.

### 4.1. Experimental setup

We evaluate the effectiveness of our adversarial attacks on OpenAI Gym MuJoCo (Todorov et al., 2012) continuous environments — *Ant*, *Hopper* and *HalfCheetah*. We evaluate the vulnerability of the victim with three attacks, including Optimal Attack (Zhang et al., 2021), Tentative Frame Attack and Optimal Frame Attack. We choose Optimal Attack as a baseline because it is the SOTA state-adversarial attacks on MuJoCo. For Optimal Attack and the victim policy, we use the pretrained models released in Zhang et al. (2021).

For Tentative Frame Attack, we first train the robust Q-function (Zhang et al., 2020) of each victim policy. Then

*Table 1.* The average reward of the victim policy (PPO) under adversarial attack on MuJoCo.

| Adversary | Ant | Hopper | HalfCheetah |
|---|---|---|---|
| epsilon | 0.15 | 0.07 | 0.15 |
| None | 5861.10 ± 609.63 | 3290.41 ± 397.13 | 7102.41 ± 121.03 |
| Optimal Attack | -493.22 ± 40.49 | 637.30 ± 3.32 | -657.60 ± 288.10 |
| Tentative Frame Attack(Ours) | -412.61 ± 58.79 | 632.70 ± 12.04 | -354.48 ± 234.38 |
| Optimal Frame Attack(Ours) | **-1240.53 ± 47.10** | **629.25 ± 13.29** | **-704.98 ± 94.28** |

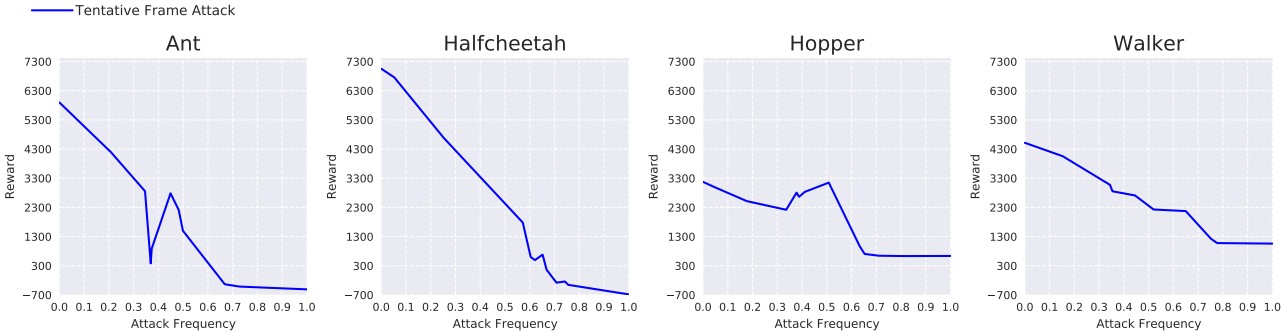

*Figure 2.* Accumulated reward (y-axis) w.r.t. portions of time steps where the agent is attacked (x-axis) by Tentative Frame Attack in 4 MuJuCo tasks. The effectiveness of attack is in inverse proportion to the accumulated reward.

we use hyper-parameter search method to properly choose $\beta$. For Optimal Frame Attack, we acquire adversarial policies w.r.t. given victim policies with the implementation in Zhang et al.'s work.

For the evaluation of attack methods, we run 50 episodes with trained victim policies and adversarial policies on every environment and report the mean and variance of accumulated reward as our experiment results.

### 4.2. Experiment results

Table 1 presents results on attacking PPO agents on MuJoCo environments such as Ant, Hopper, HalfCheetah. In all three tasks, our Tentative Frame Attack can achieve similar rewards compared with optimal attack, which attacks the victim policy at every time step. It indicates the effectiveness of choosing state by tentative functions. Besides, our Optimal Frame Attack approach markedly outperforms all other baselines which indicates our framework propose the state-of-the-art solution on deciding when to attack the victim policy.

Then, we test the vulnerability of the victim policy by using different $\beta$ and record the attack-frequency. Fig. 2 represents the attacked rewards of agents with different attack frequencies. It indicates that although Tentative Frame Attack is strong with high attack rate, it's hard to choose a proper $\beta$ with low attack rate which makes it tricky to find the best $\beta$.

## 5. Conclusion

In this paper, we proposed a strategically-timed attack in continuous control tasks named Tentative Frame Attack. We further proposed SS-MDP framework to study the properties of strategically-timed attacks on disturbing the state observations of DRL policies with a fixed preliminary adversary. Theoretical analysis shows the existence of the optimal strategically-timed attack and extensive empirical results on MuJoCo show the effectiveness of our methods. We further empirically demonstrate that strategically-timed attack is stronger than the full-time attack with fixed noise function. Currently, we haven't provide a defense method against our strategically-timed attack, which will be considered in future work.

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
