# OpenReview forum: "Strategically-timed State-Observation Attacks on Deep Reinforcement Learning Agents"
_ICML.cc/2021/Workshop/AML — ICML 2021 Workshop AML Poster_

### Official Review · Reviewer_KbFe · 2021-06-19
**This paper proposes the first Strategically-timed State-observation attack in continuous control tasks with clear modeling and impressive experiment results.**

**Rating:** Accept
**Confidence:** 3

**Review:**

Strength:
1) From the perspective of English, this paper is easy to follow.
2) The modeling is clear.
3) The attack result is impressive.
4) As the originality, the authors declare that this paper is the first work that studies the Strategically-timed State-observation attacks in continuous control tasks.

Weakness
1)	There is only one comparing method.
2)	There are too little words for the significance of this paper.

---

### Decision · Program_Chairs · 2021-06-21

**Decision:**

Accept (Poster)

**Comment:**

A good work studying the robustness of deep RL. The paper needs careful revision.

The claims made in the paper "Strategically-timed State-Observation Attacks on Deep Reinforcement Learning Agents" are unfortunately not true. Sun et al. (2020) and Yang et al. (2020) both studied this strategically timed attack in continuous control tasks. Moreover, since the authors of both of these studies experimented on the same games we can see in Figure 4 of the Sun et al. (2020) paper that the model used in [1] is superior in that it requires far fewer attacked frames to achieve a given level of score reduction.

[1] Jianwen Sun, Tianwei Zhang, Xiaofei Xie, Lei Ma, Yan Zheng, Kangjie Chen, Yang Liu: Stealthy and Efficient Adversarial Attacks against Deep Reinforcement Learning. AAAI 2020: 5883-5891.

[2] Chao-Han Huck Yang, Jun Qi, Pin-Yu Chen, Yi Ouyang, I-Te Danny Hung, Chin-Hui Lee, Xiaoli Ma: Enhanced Adversarial Strategically-Timed Attacks Against Deep Reinforcement Learning. ICASSP 2020: 3407-3411